# Additive rheology of complex granular flows

Thanh Trung Vo[1,2], Saeid Nezamabadi [2,3], Patrick Mutabaruka [2], Jean-Yves Delenne[3] & Farhang Radjai[2,4✉]

Granular flows are omnipresent in nature and industrial processes, but their rheological properties such as apparent friction and packing fraction are still elusive when inertial, cohesive and viscous interactions occur between particles in addition to frictional and elastic forces. Here we report on extensive particle dynamics simulations of such complex flows for a model granular system composed of perfectly rigid particles. We show that, when the apparent friction and packing fraction are normalized by their cohesion-dependent quasistatic values, they are governed by a single dimensionless number that, by virtue of stress additivity, accounts for all interactions. We also find that this dimensionless parameter, as a generalized inertial number, describes the texture variables such as the bond network connectivity and anisotropy. Encompassing various stress sources, this unified framework considerably simplifies and extends the modeling scope for granular dynamics, with potential applications to powder technology and natural flows.

[1] Bridge and Road Department, Danang Architecture University, Da Nang, Vietnam. [2] LMGC, University of Montpellier, CNRS, 34060 Montpellier, France. [3] IATE, UMR1208 INRAE-CIRAD-University of Montpellier-SupAgro, 34060 Montpellier, France. [4] (MSE)², UMI MIT-CNRS, 77 Massachusetts Avenue, Cambridge, MA 02139, USA. ✉email: franck.radjai@umontpellier.fr

The crucial role of granular flows in nature (landslides, debris avalanches, slope failure)[1–8] and industrial processes (handling powders and granulates, compaction, additive manufacturing)[9–13] has been at the focus cross-disciplinary research for more than thirty years[14,15]. Recent progress in theoretical understanding of granular flows has been mainly inspired by collective effects such as force chains and jamming[16–25], and by searching for relevant dimensionless control parameters[26–36].

Basic model granular media are composed of rigid particles with frictional contact interactions and, in contrast to interacting particles at the atomic scale or in colloids, rigid frictional particles are devoid of an intrinsic stress or time scale. The relevant scales are therefore set either externally, such as those arising from a confining pressure $\sigma_p$, or by collective particle motions during flow involving an inertial (or kinetic) pressure $\sigma_i \sim \rho_s \langle d \rangle^2 \dot{\gamma}^2$, where $\rho_s$ is the particle density, $\langle d \rangle$ is the mean particle diameter, and $\dot{\gamma}$ is the shear rate[27,37]. Hence, in the NPT statistical ensemble (with temperature $T = 0$ for a granular material), the apparent friction coefficient $\mu = \sigma_t/\sigma_p$, where $\sigma_t$ is the shear stress, and the packing fraction $\Phi$ are expected to be uniquely dependent on the ratio $I^2 \equiv \sigma_i/\sigma_p$, which accounts for the competing effects of particle inertia and confinement. The dimensionless number $I$ is the inertial number, defined as the ratio of two time scales (relaxation time $\langle d \rangle (\rho_s/\sigma_p)^{1/2}$ under load vs. shear time $\dot{\gamma}^{-1}$), and it was found to unify experimental and numerical data in different flow geometries[26].

Most of time, however, the particle interactions are not purely frictional and involve characteristic forces that induce additional internal stresses. A well-known example is the cohesive contact force $f_c$ in fine powders. When a powder flows, the average action of the resulting cohesive stress $\sigma_c \sim f_c/d^2$ is similar to a confining stress, tending to prevent from dilation during flow, to enhance the contact forces and to reduce the relaxation time under load[32,38,39]. As the stresses are additive, one may thus take the cohesive stress into account on the same footing as the confining pressure by replacing $\sigma_p$ by a linear combination $\sigma_n = \sigma_p + \alpha\sigma_c$, and therefore replacing $I^2$ by $I_c^2 \equiv \sigma_i/\sigma_n = I^2/(1 + \alpha\xi)$, where $\xi = \sigma_c/\sigma_p$ is the cohesion index, and $\alpha$ is a material-dependent parameter[32]. In the same way, in submerged granular flows, where the viscous drag force $\sigma_v$ is present instead of cohesive stress, $\sigma_i$ may be replaced by a linear combination $\sigma_i + \beta\sigma_v$, leading to a visco-inertial number $I_v^2 \equiv (\sigma_i + \beta\sigma_v)/\sigma_p = I^2(1 + \beta/St)$, where $St \equiv \sigma_i/\sigma_v$ is the Stokes number. In dense non-Newtonian suspensions, $I_v$ is found to be the control parameter for both $\mu$ and $\Phi$[31,35].

The above examples lead to the conjecture that granular flows are fundamentally governed by a single dimensionless parameter combining arbitrary particle interactions by virtue of stress additivity and the role of each interaction with respect to the shear rate and confining pressure. In this paper, we address this interesting issue by simulating wet granular flows such as unsaturated soils and powders at high relative humidity. The liquid bridges between particles induce both capillary and viscous (lubrication) forces whose effects on the flow behavior, together with the confining and inertial stresses, will be quantified for a broad range of parameter values, which are generally difficult to access by means of experiments.

Our results convincingly demonstrate the above conjecture not only for the apparent friction and packing fraction but also for the microstructural variables as a function of a generalized inertial number accounting for the confining, inertial, cohesive and viscous stresses. The results provide also direct evidence for the role of cohesive interactions in dense suspensions when properly interpreted in terms of effective viscosities. This work sets therefore the foundation for a unified description of complex granular flows both encompassing and extending previous work.

## Results

**Particle dynamics simulations and data collapse.** We performed extensive 3D long-shear simulations of granular samples composed of nearly 20,000 spherical particles by means of a particle dynamics method and with a broad range of the values of liquid viscosity $\eta$, surface tension $\gamma_s$, confining stress $\sigma_p$ and shear rate $\dot{\gamma}$. Our results are based on the average values of the stress tensor, velocity fields, packing fraction and granular texture in steady flow of the particles in the simulation cell between the top and bottom walls with periodic boundary conditions in the other directions. Besides repulsive elastic force and friction force, the particle interactions include the approximate analytical expressions of the capillary force $f_c$ and viscous force $f_v$ acting between neighboring particles (see Methods section). The snapshots of Fig. 1 show the boundary conditions, compressive and tensile force chains, contact and non-contact forces (due to capillary bridges) and particle velocities in steady flow.

Figure 2 displays the apparent friction coefficient $\mu$ and packing fraction $\Phi$ as a function of the inertial number $I$ for all our 281 simulations. The confining pressure $\sigma_p$ was varied in the range [15, 1000] Pa, the cohesion index $\xi$ in the range [0, 3.0] (by varying $\gamma_s$ or $\sigma_p$), the liquid viscosity $\eta$ in the range [$\eta_w$, $800\eta_w$], where $\eta_w$ is the water viscosity, and the shear rate in the wide range [0.31, 10.6]$s^{-1}$. As in dry flows[27], $\mu$ increases and $\Phi$ declines with increasing $I$ but with different values and at

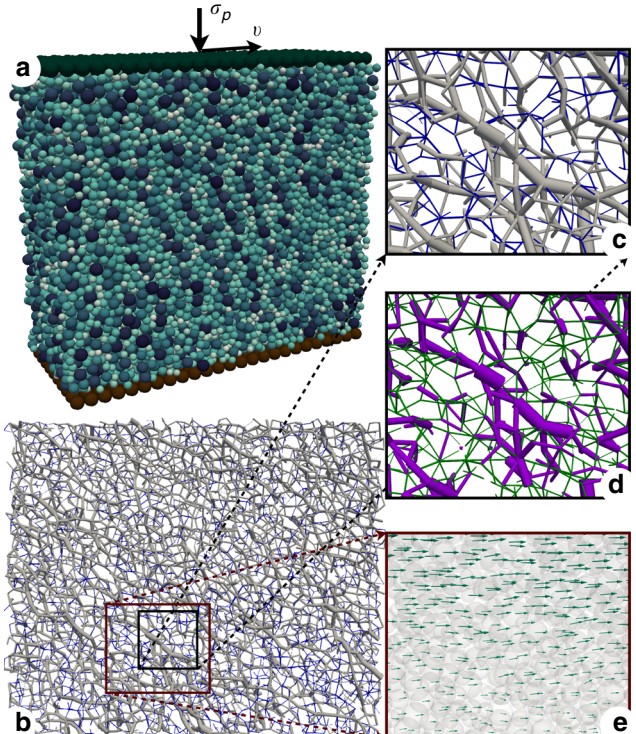

**Fig. 1 Simulated system of wet spherical particles. a** Particles in the simulation cell composed of a rough immobile bottom wall, a rough top wall subjected to a constant confining stress $\sigma_p$ and moving at constant horizontal velocity $v$, and with periodic boundary conditions along lateral directions, The particle colors are proportional to their diameters. **b** Snapshot of compressive (gray) and tensile (blue) force chains in a thin layer parallel to the flow plane. Line thickness is proportional to normal force. **c** A zoomed-in view of compressive and tensile force chains. **d** Snapshot of contact forces (violet) and non-contact capillary forces (green) in a thin window at the center of the flow plane. **e** A zoomed-in view of the particle velocity field.

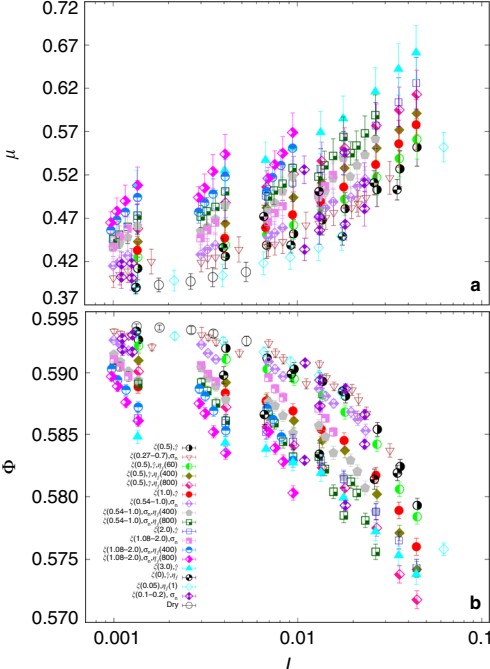

**Fig. 2 Simulation results for friction and packing fraction.** Apparent friction coefficient $\mu$ (**a**) and packing fraction $\Phi$ (**b**) as a function of the inertial number $I$ for different sets of system parameters. The data points are average values over the steady state, and the error bars represent their standard deviation in each simulation during steady-state flow. For each set of simulations, the symbols and their colors correspond to the parameters that are varied with their ranges, all other parameters being kept constant.

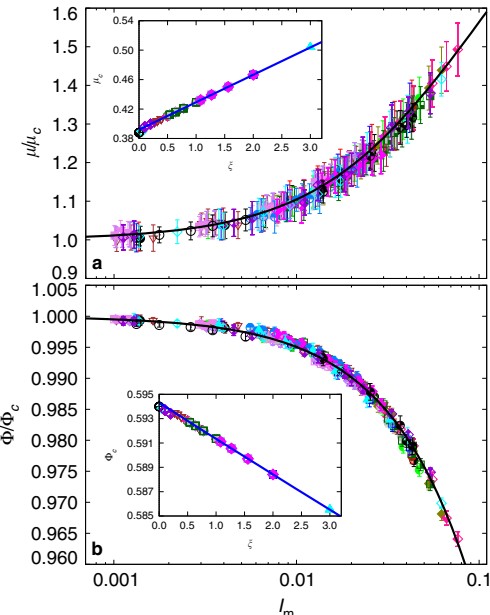

**Fig. 3 Rescaled simulation data.** Normalized apparent friction coefficient $\mu/\mu_c$ (**a**) and normalized packing fraction $\phi/\phi_c$ (**b**) as a function of the generalized inertial number $I_m$ defined by Eq. (1) with $\alpha \simeq 0.062$ and $\beta \simeq 0.075$. The error bars represent the standard deviation of the data over the steady-state flow. The symbols are the same as those in Fig. 2. The black-solid lines are the analytical expressions of Eqs. (6) and (7). The insets show the evolution of quasistatic values $\mu_c$ and $\Phi_c$ of the apparent friction coefficient and packing fraction, respectively, with the cohesion index $\xi$, and their linear fits (blue-solid lines).

different rates depending on the viscosity and cohesion index. These differences are observed at both low values (quasi-static flow) and high values (inertial flow) of $I$, and the variability of $\mu$ and $\Phi$ with the variation of $I$ is of the same order of magnitude as with the variation of viscous and cohesive parameters.

The issue is whether all these different values of apparent friction and packing fraction can be expressed as a collapsed function of a single dimensionless number combining surface tension, liquid viscosity, confining pressure and shear rate. In other words, can $I$ be replaced by a more general inertial number $I_m$ that simultaneously accounts for the capillary, viscous and inertial forces? This is indeed what we observe in Fig. 3, displaying all the data points of Fig. 2 as a function of a modified inertial number defined by

$$I_m = I\left(\frac{1 + \beta/St}{1 + \alpha\xi}\right)^{1/2}, \quad (1)$$

The values of $\mu$ and $\Phi$ are normalized by $\mu_c$ and $\Phi_c$, respectively, which design their quasi-static values ($I_m \to 0$) and vary linearly with $\xi$, as shown in the two insets to Fig. 3. The parameter values $\alpha \simeq 0.062$ and $\beta \simeq 0.075$ were determined from two series of simulations but we see that they lead to data collapse for all other simulations, including those of dry cohesionless flows. This means that $\alpha$ and $\beta$ depend only on the material parameters (particle shape and size distribution, friction coefficient between particles) and not on the cohesive and viscous interactions.

**Visco-cohesive inertial number.** The physical argument behind the definition of $I_m$ is the following. There are four characteristic stresses of different origins governing the flow: confining stress $\sigma_p$, inertial stress $\sigma_i$, viscous stress $\sigma_v$, and capillary stress $\sigma_c$. The key

variable for inertial flows is the shear rate. We thus distinguish the characteristic stresses that depend on the shear rate, i.e., $\sigma_i$ and $\sigma_v$, from those that are independent of the shear rate, i.e., $\sigma_p$ and $\sigma_c$. By virtue of stress additivity, the total shear-dependent stress is a linear combination $\sigma_i + \beta\sigma_v$ of the former, and the total shear-independent stress is a linear combination $\sigma_p + \alpha\sigma_c$ of the latter. Hence, the flow variables (apparent friction coefficient and packing fraction) are expected to depend on the ratio $(\sigma_i + \beta\sigma_v)/(\sigma_p + \alpha\sigma_c)$, which simply represents the relative magnitude of the two groups of stresses. We define the generalized inertial number $I_m$ as the square root of this ratio, leading to the expression (1) by setting $I = (\sigma_i/\sigma_p)^{1/2}$, $\xi = \sigma_c/\sigma_p$ and $St = \sigma_i/\sigma_v$.

These primary dimensionless parameters can be evaluated from the system control parameters. The order of magnitude of the viscous stress $\sigma_v$ is conveniently evaluated by replacing the average relative velocity $\vartheta \sim \dot{\gamma}\langle d\rangle$ induced by shearing in Eq. (15) and considering the dissipated power per unit volume $f_v\delta_{\text{rupt}}/\langle d\rangle^3$, where $\delta_{\text{rupt}}$ is the debonding distance, yielding $\sigma_v \sim \eta\dot{\gamma}$. The capillary stress is of the order of the capillary force at contact ($\delta = 0$) in Eq. (11) divided by the cross section $\langle d\rangle^2$: $\sigma_c \sim \gamma_s/d$). Hence, the dimensionless parameters are given by $I = \dot{\gamma}\langle d\rangle(\rho_s/\sigma_p)^{1/2}$, $\xi = \gamma_s/(\sigma_p\langle d\rangle)$, and $St = \rho_s\langle d\rangle^2\dot{\gamma}/\eta$. Note that all other dimensionless variables can be expressed in terms of these three independent parameters. For example, the capillary number is given by $Ca = \sigma_v/\sigma_c = I^2/(\xi St)$.

**Fitting functions for apparent friction and packing fraction.** According to Eq. (1), $I_m \to 0$ only if $I \to 0$. In this quasistatic limit, the flow variables may still depend on $\xi$, which is the only dimensionless variable in the absence of inertial and viscous stresses. Accordingly, the rheology is expected to be described by

the following general equations:

$$\mu = \mu_c(\xi)\, G_\mu(I_m), \tag{2}$$

$$\Phi = \Phi_c(\xi)\, G_\Phi(I_m), \tag{3}$$

These equations are based on functional distinction between the quasistatic limit ($I_m \to 0$) and shear-rate dependent behavior through $I_m$. When $I_m \to 0$, we have $\mu \to \mu_c$ and $\Phi \to \Phi_c$, and thus $G_\mu \to 1$ and $G_\Phi \to 1$. According to the simulation data displayed in Fig. 3, $\mu_c$ and $\Phi_c$ are linear functions of $\xi$:

$$\mu_c \simeq \mu_0(1 + a\xi) \tag{4}$$

$$\Phi_c \simeq \Phi_0(1 - b\xi) \tag{5}$$

with $a \simeq 0.095$ and $b \simeq 0.005$. The limit values $\mu_0 \simeq 0.392$ and $\Phi_0 \simeq 0.594$ are the values of the apparent friction coefficient and packing fraction in the absence of cohesive and viscous forces (dry limit), respectively. Remarkably, the limit value $\Phi_0 \simeq 0.594$ obtained here by simulations is equal to the measured value of packing fraction in glass-bead flows[29].

The data points are nicely fitted by the following functional forms:

$$\frac{\mu}{\mu_c} = G_\mu = 1 + \frac{\Delta_\mu}{1 + I_\mu/I_m} \tag{6}$$

$$\frac{\Phi}{\Phi_c} = G_\Phi = \frac{1}{1 + I_m/I_\Phi} \tag{7}$$

with $\Delta_\mu \simeq 1.100$, $I_\mu \simeq 0.095$ and $I_\Phi \simeq 2.010$. Interestingly, these functions are the same as those previously used for dry granular flows, with the new *visco-cohesive* inertial parameter $I_m$ replacing $I$[28,40]. This shows that, up to the values of $I_\mu$, $I_\Phi$ and $\Delta_\mu$, our simulation data are consistent with the experimental measurements of ref. [40] and ref. [28] in the dry case. The values of $I_\mu$ and $I_\Phi$ are also very close to those obtained in the simulations of Roy et al.[34] in a ring shear cell once re-expressed in terms of our definitions. While the functional forms are general[28,31,32,34,35], the fitting parameters depend on the space dimension and material properties of the granular materials such as particle size distributions, particle shape and friction coefficient between particles. The values of $\alpha$ and $\beta$ reflect the relative roles of viscous, inertial and cohesive forces in collective dissipation mechanisms whereas the values of $I_\mu$, $I_\Phi$, and $\Delta_\mu$ account for the effects of material parameters. Note also that, given the investigated range of values of $I_m$, Eq. (7) can be linearized with an error $\sim 10^{-3}$, in agreement with refs. [32,34,41].

The fitting forms reveal the double role played by cohesion. Since $I_m$ is a decreasing function of $\xi$, $G_\mu$ declines with increasing $\xi$ (dynamic effect) whereas $\mu_c$ increases (quasistatic effect). We can easily check from the parameter values that the quasistatic effect prevails although the dynamic effect becomes important at large values of $\xi$ and $I_m$. These roles are inversed for the packing fraction: $\Phi_c$ declines whereas $G_\Phi$ increases when $\xi$ is increased. In other words, the cohesive interactions lead to lower packing fraction (enhanced dilatancy due to cohesive forces) but the inertial effects tend to increase it.

**Transition to the NVT ensemble and effective viscosities.** The flow behavior can alternatively be described in the NVT ensemble (constant-volume shearing) in terms of effective normal and shear viscosities $\eta_n$ and $\eta_t$ defined by $\sigma_n = \eta_n \dot\gamma$ and $\sigma_t = \eta_t \dot\gamma$, respectively, where $\sigma_t = \mu\sigma_n$ is the shear stress[29]. In this ensemble, the packing fraction $\Phi$ replaces pressure $\sigma_n$ as control parameter, and the rheology is characterized by the functions $\eta_n(\Phi)$ and $\eta_t(\Phi)$[29]. This is the approach mostly used in experiments on

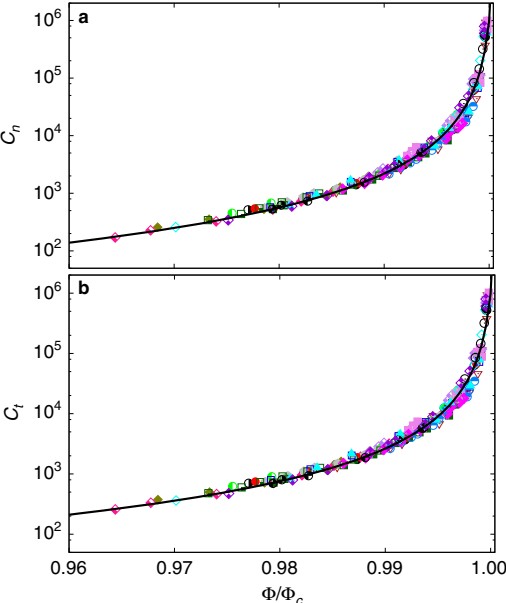

**Fig. 4 Effective viscosities in the NVT ensemble.** Normalized bulk viscosity $c_n$ (**a**) and shear viscosity $c_t$ (**b**) as a function of the normalized packing fraction $\Phi/\Phi_c$, where $\Phi_c$ is the quasistatic packing fraction depending linearly on the cohesion index; see Eq. (5). The error bars represent the standard variation over steady-state flow, and the symbols and their colors are the same as those in Fig. 2. The solid lines are the functional forms of Eq. (8) and Eq. (9), respectively.

suspensions. Although our simulations were carried out under NPT conditions, we may deduce $\eta_n(\Phi)$ and $\eta_t(\Phi)$ in the NVT ensemble from $\mu(I_m)$ and $\Phi(I_m)$. Since no external stress is imposed in NVT, the shear stress $\sigma_t$ is a dynamic variable that should scale with the internal shear-dependent stress $\sigma_i + \beta\sigma_v$. Moreover, the NPT and NVT points of view should be compared at the same normal stress state, i.e., $\sigma_n = \sigma_p + \alpha\sigma_c$. Hence, according to Eq. (1), $\sigma_n = c_n(\sigma_i + \beta\sigma_v) \equiv c_n\sigma_n I_m^2 = \eta_n\dot\gamma$, implying $c_n = 1/I_m^2 = \eta_n/(\beta\eta + \rho_s\langle d\rangle^2\dot\gamma)$, and $\sigma_t = c_t(\sigma_i + \beta\sigma_v)$ with $\sigma_t = \mu/I_m^2 = \eta_t/(\beta\eta + \rho_s\langle d\rangle^2\dot\gamma)$. In this way, in a volume-controlled flow, $c_n$ and $c_t$ represent dimensionless viscosities with $\beta\eta + \rho_s\langle d\rangle^2\dot\gamma$ as reference viscosity; see Supplementary Note 3 for more detail.

Figure 4 displays the effective dimensionless viscosities as a function of $\Phi$. We see that all the data points collapse on a master curve when $\Phi$ is normalized by the critical packing fraction $\Phi_c$. Both viscosities diverge as $\Phi \to \Phi_c$ and they are nicely fitted by the analytic expressions

$$c_n = \frac{1}{I_\Phi^2}\left(\frac{\Phi}{\Phi_c - \Phi}\right)^2, \tag{8}$$

$$c_t = \mu c_n = \frac{1}{I_\Phi^2}\left(\frac{\Phi}{\Phi_c - \Phi}\right)^2\left\{1 + \frac{\Delta_\mu}{1 + \frac{I_\mu}{I_\Phi}\frac{\Phi}{\Phi_c - \Phi}}\right\}, \tag{9}$$

readily deduced from the expressions of $c_n$ and $c_t$ as a function of $I_m$ together with Eqs. (6) and (7). As in suspensions, $1/c_t = I_m^2/\mu$ represents a generalized fluidity parameter of granular flows[42].

**Scaling of coordination number and bond anisotropy.** Although $I_m$ provides a unified description of the rheology by capturing the effects of particle interactions on the apparent friction coefficient and packing fraction and, alternatively, the effective viscosities, it is essential to check its robustness with

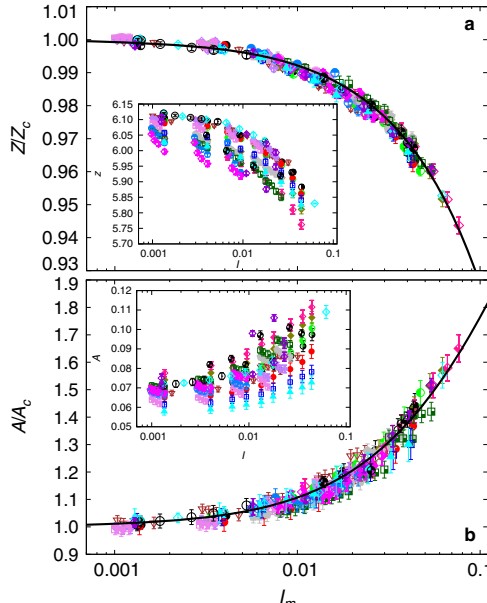

**Fig. 5 Scaling of microstructural variables.** Number density $Z$ of capillary bonds (**a**) and bond orientation anisotropy $A$ (**b**) as a function of the generalized inertial number $I_m$ defined by Eq. (1) with $\alpha \simeq 0.062$ and $\beta \simeq 0.075$, for sets of parameters. The error bars represent the standard variation over steady-state flow, and the symbols and their colors are the same as those in Fig. 2. The solid lines are fitting forms similar to those used for $\mu/\mu_c$ and $\Phi/\Phi_c$ with different parameter values; see Supplementary Note 2.

respect to microstructural variables. Let us consider the shear plane defined by the directions of shearing $\dot{\gamma}$ and confining stress $\sigma_p$ (the flow being translationally invariant in the lateral direction). In this plane, the number density $E(\Theta)$ of bonds (both contacts and non-contact capillary bridges) per particle along a direction $\Theta$ can be approximated by[15,21,43]

$$E(\Theta) = \frac{Z}{2\pi}\{1 + A\cos 2(\Theta - \Theta_b)\}, \tag{10}$$

which represents a truncated Fourier transform of $E(\Theta)$ with a period $\pi$ since no intrinsic polarity can be attributed to contact orientations. Higher-order terms are generally negligibly small in granular flows. The coordination number $Z$ and bond orientation anisotropy $A$ are the lowest-order descriptors of granular texture. The angle $\Theta_b$ is the privileged bond orientation, and its value is close to $\pi/4$ with respect to the flow direction.

Figure 5 shows $Z$ and $A$ normalized by their quasistatic values $Z_c$ and $A_c$, respectively, as a function of $I_m$. We see that, despite their larger variability as compared to $\mu(I_m)$ and $\Phi(I_m)$, the data points collapse on a master curve within our statistical precision as a function of $I_m$ with the same values $\alpha \simeq 0.062$ and $\beta \simeq 0.075$ as in $\mu(I_m)$ and $\Phi(I_m)$. The insets show the extent to which the same data points are dispersed when plotted as a function of $I$. We find that $Z_c$ is a decreasing linear function of $\xi$ (as $\Phi_c$) and $A_c$ is an increasing linear function of $\xi$ (as $\mu_c$); see Supplementary Fig. 3. Moreover, the functional forms that fit $A/A_c$ and $Z/Z_c$ are the same as $G_\mu$ and $G_\Phi$ as a function of $I_m$ with different values of their free fitting parameters; see Supplementary Eqs. (3) and (4). This scaling of microstructural variables with $I_m$ through a functional dependence similar to flow variables clearly indicates that the shear strength is mainly due to the increasing aptitude of the particles to self-organize in an anisotropic network[41,44].

## Discussion

The additive rheology of granular materials, as demonstrated in this paper, is by no means self-evident. The expectation that a system involving several dimensionless parameters can ultimately be described by a single parameter combining those parameters is unusual. The deep reason behind such a behavior is the very nature of granular materials in which the particle interactions are concentrated at the contact points and the local dynamics is controlled by the shear rate. Hence, by careful distinction of stresses depending on the shear rate (first group) from those that are independent (second group), a single parameter can be defined by means of the stress additivity property. In this respect, the modified inertial number $I_m$ is a conceptual extension of the inertial number to arbitrary interactions between particles. Such a scaling works, however, by distinguishing the quasistatic limit and normalizing the packing fraction, apparent friction coefficient, coordination number and bond orientation anisotropy by their quasistatic limit values that depend on the dimensionless numbers of the second group ($\xi$ in our case). Encompassing dry and cohesive-viscous flows, quasistatic and dynamic states, flow variables and microstructural parameters, this scaling provides a general framework for complex granular flows.

This framework can be applied to quantify the effects of friction coefficient between particles and particle shape and size distributions as material parameters that can influence the relative roles of internal stresses in the collective flow behavior, and thus the fitting parameters. Previous results on the rheology of granular materials suggest that the functional forms fitting the master curves should not depend on the particle shapes, size distributions and friction coefficient, although their free parameters will certainly do[45–52]. This is, however, a crucial step forward for application to different types of granular materials and for comparison with experiments.

It is also useful to point out that, for given particle size $d$ and density $\rho_s$, each characteristic stress $\Sigma$ corresponds also to a characteristic time $T = d(\rho_s/\Sigma)^{1/2}$. This relation implies that, by virtue of stress additivity, the inverse quadratic times $1/T^2$ are additive. Hence, using the characteristic times and this rule, we arrive at the same expression of $I_m$ as in Eq. (1). Another interesting issue concerns an alternative rheology based on a 'multiplicative' expression of the flow variables, rather than additive combination of control parameters, as in ref. [34] applied to a cohesive-frictional granular material. In this approach, the apparent friction coefficient $\mu$ is expressed as a product of distinct functions of the primary dimensionless control parameters. This multiplicative partition works quite well for the cohesion index, and a prefactor similar to $\mu_0(1 + a\xi)$ in Eq. (4) is obtained. For the other functions, it is worth considering in more detail how they relate to the framework developed in this paper.

Let us finally recall that the inertial number was initially introduced in the context of cohesionless granular materials where the gravity or applied confining stress prevail. However, the cohesive stress may largely exceed the confining stress in fine powders, and therefore the inertial and viscous stresses must be advantageously compared to the cohesive stress rather than the confining stress. It is easy to see that, in this limit ($\sigma_p \to 0$), the modified inertial number is reduced to $I_m = \{Ca(\beta + St)/\alpha\}^{1/2}$. This is a simple expression that is expected to scale cohesive processes such as wet granulation and impact dynamics of cohesive aggregates. We thus propose to use impact experiments as a convenient means to investigate this scaling.

## Methods
**The model of capillary bridge.** We assume that the liquid inside the agglomerate is in the 'pendular' state with a uniform distribution of capillary bridges between particles[38,53–59]. This distribution may be a consequence of mixing the liquid with

the particles, drainage of a saturated packing, or capillary condensation from a vapor. For a separation distance above a debonding distance $d_{\text{rupt}}$, the bridge breaks and its liquid is shared between the two particles proportionally to their sizes[53,60].

The capillary force $f_c$ between two particles depends on the liquid volume $V_b$ of the bond, liquid-vapor surface tension $\gamma_s$ and particle-liquid-gas contact angle $\theta$. We used the following expression[61]:

$$f_c = \begin{cases} -\kappa R, & \text{for } \delta_n < 0, \\ -\kappa R\, e^{-\delta_n/\lambda}, & \text{for } 0 \leq \delta_n \leq d_{\text{rupt}}, \\ 0, & \text{for } \delta_n > d_{\text{rupt}}, \end{cases} \quad (11)$$

where $R = \sqrt{R_i R_j}$ is the geometrical mean radius and the pre-factor $\kappa$ is

$$\kappa = 2\pi\gamma_s \cos\theta. \quad (12)$$

The debonding distance $d_{\text{rupt}}$ is given by[53]

$$d_{\text{rupt}} = \left(1 + \frac{\theta}{2}\right) V_b^{1/3}. \quad (13)$$

The characteristic length $\lambda$ in Eq. (11) is given by

$$\lambda = c\, h(r) \left(\frac{V_b}{R'}\right)^{1/2}, \quad (14)$$

where $R' = 2R_i R_j/(R_i + R_j)$ and $r = \max\{R_i/R_j; R_j/R_i\}$ are the harmonic mean radius and the size ratio between two particles, $h(r) = r^{-1/2}$, and $c \simeq 0.9$. In all simulations, we set $\theta = 0$ with the assumption that the particles are covered by a layer of the wetting liquid.

Note that the simulated system is an idealized model of wet granular materials in the pendular state. Nevertheless, we believe that our results can be extended to higher amounts of liquid since the liquid can easily flow in an unsaturated material to wet larger particle areas with a lower Laplace pressure. This leads to a nearly constant cohesive stress as far as the material is not fully saturated[54,55]. Hence, the leading effect of increased liquid volume is simply the increase of debonding distance when Eq. (11) is used.

**The normal viscous force**. The normal lubrication force $f_{\text{vis}}$ due to the effect of liquid bridges between two smooth spherical particles is given by[62–64]

$$f_{\text{vis}} = \frac{3}{2}\pi R^2 \eta \frac{v_n}{\delta_n}, \quad (15)$$

where $\eta$ is the liquid viscosity and $v_n$ is the relative normal velocity, assumed to be positive when the gap $\delta_n$ is decreasing. This force diverges when the gap $\delta_n$ tends to zero. But for slightly rough particles, the characteristic size of the asperities allows for collision in finite time. Hence, we introduce a characteristic length $\delta_{n0}$ corresponding to the size of asperities so that the lubrication force for $\delta_n > 0$ is given by

$$f_{\text{vis}} = \frac{3}{2}\pi R^2 \eta \frac{v_n}{\delta_n + \delta_{n0}} \quad \text{for } \delta_n > 0 \quad (16)$$

For $\delta_n < 0$ (a contact between two particles), we assume that the lubrication force remains equal to its largest value:

$$f_{\text{vis}} = \frac{3}{2}\pi R^2 \eta \frac{v_n}{\delta_{n0}} \quad \text{for } \delta_n \leq 0. \quad (17)$$

In our simulations, we set $\delta_{n0} = 5 \times 10^{-4} d_{\text{min}}$, where $d_{\text{min}}$ is the smallest particle diameter. This value is sufficiently small to allow the lubrication force to be effective without leading to its divergence at contact.

**Simulation method**. For the simulations, we used the molecular dynamics (MD) method with frictional contact interactions modeled by linear elastic repulsion along the normal direction and linear spring with a Coulomb threshold along the tangential direction, together with the previous approximate expressions of capillary force and viscous force acting between neighboring particles.

The particle displacements are calculated by step-wise resolution of Newton's second law:

$$m_i \frac{d^2 \mathbf{r}_i}{dt^2} = \sum_j [(f_n^{ij} + f_c^{ij} + f_{\text{vis}}^{ij})\mathbf{n}^{ij} + f_t^{ij}\mathbf{t}^{ij}],$$

$$\mathbf{I}_i \frac{d\boldsymbol{\omega}_i}{dt} = \sum_j f_t^{ij}\mathbf{c}^{ij} \times \mathbf{t}^{ij}, \quad (18)$$

where particle $i$ is assumed to interact with its neighbors $j$ via normal contact forces $f_n$, tangential contact forces $f_t$, capillary forces $f_c$, and viscous forces $f_{\text{vis}}$. $\boldsymbol{\omega}_i$ is the rotation vector of particle $i$, and $m_i$, $\mathbf{I}_i$, and $\mathbf{r}_i$ are its mass, inertia matrix, and position, respectively. $\mathbf{n}^{ij}$ denotes the unit vector perpendicular to the contact plane between the particles $i$ and $j$ and points from $j$ to $i$. $\mathbf{t}^{ij}$ is the unit vector in the contact plane pointing in the direction opposite to the relative tangential displacement of the two particles. $\mathbf{c}^{ij}$ is the vector joining the center of particle $i$ to the contact point with particle $j$.

**Table 1 Constant simulation parameters.**

| Parameter | Symbol | Value | Unit |
|---|---|---|---|
| Number of particles | $N_p$ | 19,628 | |
| Smallest particle diameter | $d_{min}$ | 800 | μm |
| Particle density | $\rho_s$ | 2600 | kg.m$^{-3}$ |
| Friction coefficient | $\mu_s$ | 0.4 | |
| Normal stiffness | $k_n$ | $10^6$ | N/m |
| Tangential stiffness | $k_t$ | $8.10^5$ | N/m |
| Normal damping | $\gamma_n$ | 0.5 | Ns/m |
| Tangential damping | $\gamma_t$ | 0.5 | Ns/m |
| Contact angle | $\theta$ | 0 | degree |
| Time step | $\delta t$ | $3.10^{-7}$ | s |

The normal contact force $f_n$ is the sum of four contributions:

$$f_n = f_n^e + f_n^d + f_c + f_{\text{vis}}, \quad (19)$$

where $f_n^e$ is the elastic repulsion force, and $f_n^d$ is the normal damping force. The elastic force $f_n^e = -k_n\delta_n$ is a linear function of the normal elastic deflection $\delta_n$, where $k_n$ is the normal stiffness, and the damping force $f_n^d = \gamma_n\dot{\delta}_n$ is proportional to the relative normal velocity $\dot{\delta}_n$, where $\gamma_n$ is the normal viscous damping parameter. These elastic and damping forces occur only when two particles are in contact ($\delta_n < 0$). The tangential force $f_t$ is composed of an elastic force $f_t^e = -k_t\delta_t$ and a damping force $f_t^d = \gamma_t\dot{\delta}_t$, where $k_t$ is the tangential stiffness, $\gamma_t$ is the tangential damping parameter, and $\delta_t$ and $\dot{\delta}_t$ are the tangential displacement and velocity, respectively. According to the Coulomb friction law, the tangential force is below $\mu f_n$, where $\mu$ is the friction coefficient[65–67]:

$$f_t = -\min\left\{\left| -k_t\delta_t + \gamma_t\dot{\delta}_t\right|, |\mu f_n|\right\} sgn(\dot{\delta}_t). \quad (20)$$

The tangential lubrication force was neglected as it is one order of magnitude below the normal lubrication force.

The equations of motion were integrated by a step-wise velocity-Verlet aglorithm[68]. The constant physical parameters were set to typical values of fine granular materials composed of hard particles. We used a weak size polydispersity with a uniform distribution of particle volumes and a ratio 2 between the largest and smallest particle diameters. All the constant physical and numerical parameter values are given in Table 1. Note that the relative pressure $p^* = \sigma_p/(\langle d\rangle k_n)$, representing the ratio of contact deflection to particle diameter, is of the order of $10^{-5}$. Our simulations correspond therefore with a high precision to the ideal limit of perfectly rigid particles.

## Data availability
All relevant data are available upon request from the authors.

## Code availability
The simulation code is available at sourcesup.renater.fr/www/cfgd3d and upon reasonable request from the authors.

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

## Acknowledgements

The authors acknowledge financial support by the Ministry of Education and Training in Vietnam, Campus France, and Labex NUMEV (University of Montpellier). This work was realized with the support of MESO@LR-Platform at the University of Montpellier.

## Author contributions

F.R., S.N. and J.Y.D. conceived the research idea. P.M. developed the particle dynamics program CFGD++3D used in this work and provided support for the simulations. T.T.V. carried out all the simulations, analyzed the data, and prepared all the figures. All authors discussed the results and contributed to the writing of the paper.

## Competing interests

The authors declare no competing interests.

**Additional information**

