## [Peer Review File · Nature Communications]

Editorial Note: Parts of this Peer Review File have been redacted as Reviewer #2 cannot grant Nature Communications permission to publish his/her report.

Reviewers' comments:

Reviewer #1 (Remarks to the Author):

I have now reviewed the submitted manuscript "Complex granular flows are simple" by Thanh Trung Vo et. al. This manuscript presents an extensive simulation of binary spherical packing systems where the shear rate is a variable and the authors claim that the rheology of the system can be described by a single dimensionless parameter largely due the additive property of the stress. The manuscript is well-written, and it is fairly clear, and the claim is supported through a series of universality observed in both local (e.g. coordination number) and non-local (e.g. packing fraction etc.) granular parameters.

While this is an interesting work and certainly of interest to the granular material community, it is limited in the sense that: 1. It is a simulation ONLY work with no support of experimental validation, 2. The system is a binary spherical packing and the claims of the paper remain to be seen in other granular systems, 3. Interesting localised phenomena observed in granular systems such as shear banding and/or shear thinning in systems with viscosity is not seen or reported in this manuscript, 4. The role of the fabric of granular materials and shape is not investigated. It is not clear if the universality shown in the manuscript is held when particle shapes deviate from spherical and/or the particle size distribution varies.

I do not think that this work is general enough for publication in Nature Communications. This is a very specific work and a specialised journal such as Granular Matter or Journal of Mechanics and Physics of Solids will be better platforms for this paper.

Some notes to the authors; the bibliography is quite biased in that they extensively cite their own papers. There is a substantial amount of work done in this area and this manuscript fails to include them in the biblio.

Also, I presume the authors mean to say "Competing interest" in the section "Additional information" instead of "Completing interest".

Reviewer #2 (Remarks to the Author):

[Redacted]

Reviewer #3 (Remarks to the Author):

Vo and coworkers study the rheological properties of wet granular materials via molecular dynamics simulations. They find that the stress additivity leads to a dimensionless parameter I_m , which controls the rheology of granular materials with all kinds of interactions including friction, capillary force, and viscous force. Taking I_m as the control parameter, they find that quantities like apparent friction coefficient, packing fraction, number density of capillary bonds, and bond anisotropy all show nice scaling collapse.

I find this work very interesting. It is really nice to figure out the generalized inertial number I_m and

unify the rheology of granular materials which had to be described by multiple dimensionless parameters previously. I believe this is an important advancement of the field if the results are indeed general. I am inclined to recommend publication. But before that, I hope the authors could consider the following comments and make necessary changes to the manuscript.

-- There are quite a few tunable parameters. Although it is usually impracticable to access an enough wide range of all parameters, compared with other parameters which cover at least one order of magnitude, the confining pressure only varies from 106.3 to 276.4 Pa. Is there any special reason to choose such a narrow pressure window? Since this is simulation work, I would like to see whether the results remain the same when pressure is varied over orders of magnitude. For me, this is crucial to check the generality of the work. Φ_0 in Eq. (5) would definitely change if pressure varies a lot, so would μ_0 in Eq. (4) I guess. Will such changes affect the scaling functions?

-- More information about the simulation methods should be provided. It's better to explicitly show the equations of motion. What is the form of the particle-particle interaction, harmonic or Hertzian? Is the shear applied by just moving the top wall horizontally, without the constraint of shear gradient such as using SLLOD?

-- Minor points: In the first paragraph, " τ " in the expression of friction coefficient is not defined. In the first paragraph of page 4, "In a similar vein, ..., with σ_t ", " σ " should be " c ". In the sentence " δ is the gap length" below Eq. (12), should " δ " be " δ^{ij} "?

Reviewer #1

C- I have now reviewed the submitted manuscript “Complex granular flows are simple” by Thanh Trung Vo et. al. This manuscript presents an extensive simulation of binary spherical packing systems where the shear rate is a variable and the authors claim that the rheology of the system can be described by a single dimensionless parameter largely due the additive property of the stress. The manuscript is well-written, and it is fairly clear, and the claim is supported through a series of universality observed in both local (e.g. coordination number) and non-local (e.g. packing fraction etc.) granular parameters.

While this is an interesting work and certainly of interest to the granular material community, it is limited in the sense that: 1. It is a simulation ONLY work with no support of experimental validation, 2. The system is a binary spherical packing and the claims of the paper remain to be seen in other granular systems, 3. Interesting localized phenomena observed in granular systems such as shear banding and/or shear thinning in systems with viscosity is not seen or reported in this manuscript, 4. The role of the fabric of granular materials and shape is not investigated. It is not clear if the universality shown in the manuscript is held when particle shapes deviate from spherical and/or the particle size distribution varies.

R- We are thankful for your general positive view of this work, of particular interest to the community of granular materials. This work was made possible by using extensive 3D simulations involving the variation of several parameters in a broad range of values. Clearly, such a vast parametric investigation cannot be presently accessed by experiments. For partial validation, we do have provided several hints from the literature. First, the functional form that we use for fitting the data points were obtained from experiments as mentioned below the equations (6) and (7):

« Interestingly, these functions are strictly the same as those previously used for dry granular flows, with the new *visco-cohesive* inertial parameter I_m replacing I [40, 28] ». The references [40] and [28] are experimental works. This is a strong hint that the numerical results of the present work reproduce the same trends as those observed in experiments. To clarify this point, we added the following sentence to the paragraph following equations (6) and (7):

« This shows that, up to the values of I_μ , $I\Phi$ and $\Delta\mu$, our simulation data are consistent with the experimental measurements of Refs. [40] and [28] in the dry case. The values of I_μ and $I\Phi$ are also very close to those obtained in the simulations of Roy et al. [34] in a ring shear cell once reexpressed in terms of our definitions of the parameters. »

To give even more hints for comparison with experiments, we also added the following sentence:

« Remarkably, the limit value $\Phi_0 \approx 0.594$ obtained here by simulations is equal to the measured value of packing fraction in experiments on glass-bead flows [29]. »

Furthermore, it is important also that a simulation work is able to suggest new experiments. This is in fact the last suggestion of our paper: « This is a simple expression that is expected to scale cohesive processes such as wet granulation and impact dynamics of cohesive aggregates. We thus propose to use impact experiments as a convenient means to investigate this scaling. ».

Finally, the reviewer certainly agrees with us that this work is not just a compilation of raw data from numerical simulations. These simulations were actually designed to be used as « numerical experiments » to check a theoretical idea. This was even the point of departure of this work and its main motivation as put on the first page: « The above examples lead to the conjecture that granular flows are fundamentally governed by a single dimensionless parameter combining arbitrary particle interactions by virtue of stress additivity. In this paper, we address this interesting issue by simulating wet granular flows... »

Regarding your second point, the system is not a « binary » granular packing. The particle diameters are uniformly distributed by volume fractions between two values. This was mentioned in the last paragraph of the Method section: « We used a weak size polydispersity with a uniform distribution of particle volumes and a ratio of two between the largest and smallest particle diameters. »

Concerning your third point, the boundary conditions were defined so that no shear banding occurs. If shear-banding occurred, the system would have not been appropriate for the rheological investigation undertaken in this paper. The roughness of the walls is a key parameter. We understand that the simulation conditions should have been given in more detail. In the revised version, we added a Supplemental Material that includes the velocity profiles. The issue of strain localization merits

certainly to be investigated by a stability analysis of the general rheology introduced in this paper.

Regarding particle shape, we agree with the reviewer that it is an important material parameter of granular materials. The systematic investigation of particle shape effects requires experimental and numerical tools and methods that were not available until very recently to researchers. Some of the authors of this work have extensively worked on this topic. However, most of the work on the rheology of granular materials is led with a *reference material*, namely a weakly polydisperse packing of spherical particles (glass beads,...). Several studies show that various particle shapes lead to the same rheological behavior but with different values of the rheological parameters. For example, in our simulations the values of the packing fraction for polyhedral particles would have been lower, but the trends as a function of the control parameters of our system are expected to be similar. A general rheology, such as the one presented in our paper, provides a reference behavior from which possible deviations due to material parameters (particle shape, size distribution, interparticle coefficient of friction) can be quantified. This is a vast prospect that requires the efforts of the whole community in the future. We never claimed to be as « universal » (we never used this word) as assumed by the reviewer. However, we believe that this work provides a pretty general description of the rheology of granular materials for a reference material. What is more, this generalization concerns cohesive granular materials, which are presently at the focus of several communities because of their importance for most powders and fine soils. Put differently, there are two groups of parameters that characterize the material variability of granular materials: 1) particle properties (shape, size distributions, strength...), and 2) particle interactions (friction, capillary cohesion, capillary viscosity, elasto-plastic contact behavior...). In this paper, we are interested in the second group, which still raise fundamental issues as those considered in this work. As an action in response to your suggestion, we did several editorial changes, some of which are mentioned below:

In the paragraph following equations (6) and (7): « While the functional forms are general [28, 31, 32, 35, 34], the fitting parameters depend on the space dimension and material properties of the granular materials such as particle size distributions, particle shape and friction coefficient between particles. The values of α and β reflect the relative roles of viscous, inertial and cohesive forces in collective dissipation mechanisms whereas the values of I_μ , $I\Phi$ and $\Delta\mu$ account for the effects of material parameters. »

In the abstract: « Relying on extensive particle dynamics simulations of a model granular system, we show that such complex flows of perfectly rigid particles are governed... »

In Discussion: « This framework can be applied to quantify the effects of friction coefficient between particles and particle shape and size distributions as material parameters that can influence the relative roles of internal stresses in the collective flow behavior, and thus the fitting parameters. This is a crucial step forward for

application to different types of granular materials and for comparison with experiments. »

C- I do not think that this work is general enough for publication in Nature Communications. This is a very specific work and a specialized journal such as Granular Matter or Journal of Mechanics and Physics of Solids will be better platforms for this paper.

R- We do not claim full generality although we introduce in this paper a quite broad generalization of the rheology of frictional-cohesive-viscous granular materials. We think that the reviewer also agrees that there are several papers on granular materials in Nature Communications, which are much more specific than ours. All the references 24, 25 and 28 concern model granular materials. Our work represents a breakthrough in this field touching several communities (from physics and mechanics to engineering, with applications to soils, powders, granulates....), and, in this respect, it will be of interest to a general readership.

[Redacted]

Reviewer #3

C- Vo and coworkers study the rheological properties of wet granular materials via molecular dynamics simulations. They find that the stress additivity leads to a dimensionless parameter I_m , which controls the rheology of granular materials with all kinds of interactions including friction, capillary force, and viscous force. Taking I_m as the control parameter, they find that quantities like apparent friction coefficient, packing fraction, number density of capillary bonds, and bond anisotropy all show nice scaling collapse.

I find this work very interesting. It is really nice to figure out the generalized inertial number I_m and unify the rheology of granular materials which had to be described by multiple dimensionless parameters previously. I believe this is an important advancement of the field if the results are indeed general. I am inclined to recommend publication. But before that, I hope the authors could consider the following comments and make necessary changes to the manuscript.

R- We are very thankful to the reviewer for his/her very positive appreciation of this paper with helpful comments that we used to improve the manuscript.

C- There are quite a few tunable parameters. Although it is usually impracticable to access an enough wide range of all parameters, compared with other parameters which cover at least one order of magnitude, the confining pressure only varies from 106.3 to 276.4 Pa. Is there any special reason to choose such a narrow pressure window? Since this is simulation work, I would like to see whether the results remain the same when pressure is varied over orders of magnitude. For me, this is crucial to check the generality of the work. Φ_0 in Eq. (5) would definitely change if pressure varies a lot, so would μ_0 in Eq. (4) I guess. Will such changes affect the scaling functions?

R- The range of values of the confining pressure was indeed narrower than that of other control parameters. We performed extra simulations (33 more simulations by including also dry cohesionless samples suggested by the second reviewer) with the confining pressure varying now from 15 Pa to 1000 Pa. The new data points are included in the new figures. These did not change in any way our fitting functions and more specifically the values of Φ_0 or μ_0 . As also discussed in reply to reviewer #2, the simulation parameters are such that the particles are definitely hard with an approximate value of overlap between particles normalized by particle diameter for the largest pressure $p^* = pd/k_n = 10^{-5}$, which is a tiny fraction. Clearly, if we increase the pressure by orders of magnitude, p^* will increase and at some point the packing fraction will also increase as a result of overlaps. However, such an increase in overlaps is not desirable in the MD method, which is based on rigid-particle dynamics. The effect of particle softness has been partially investigated in Ref. [34]. We added « perfectly rigid particles » to the abstract to underline this hard-particle nature of the simulations.

C- More information about the simulation methods should be provided. It's better to explicitly show the equations of motion. What is the form of the particle-particle interaction, harmonic or Hertzian? Is the shear applied by just moving the top wall horizontally, without the constraint of shear gradient such as using SLLOD?

R- We added a Supplemental Material in which more details about the method are included. The contacts in our simulations are linear. The shearing is induced by just moving the top wall.

C- Minor points: In the first paragraph, "tau" in the expression of friction coefficient is not defined. In the first paragraph of page 4, "In a similar vein, ..., with σ_t ...", "sigma" should be "c". In the sentence "delta is the gap length" below Eq. (12), should "delta" be " δ^{ij} "?

R- Indeed. We defined tau (replaced by σ_t), replaced sigma by σ_c , and replaced the symbol δ by δ^{ij} .

REVIEWERS' COMMENTS:

Reviewer #1 (Remarks to the Author):

I reviewed the Authors' response to my and other Reviewer's comments and I now have a better understanding of the work and their contribution.

I am satisfied with Authors' comprehensive response and I fully recommend this work for publication in Nature Communications. It's a significant contribution to granular dynamics.

Reviewer #2 (Remarks to the Author):

[Redacted]

Reviewer #3 (Remarks to the Author):

I am satisfied with the authors' response to my comments and their changes to the manuscript. I would now like to recommend publication as is.

Replies to reviewers

We thank all reviewers for their helpful and suggestive comments on this work and recommending this manuscript for publication in Nature Communications.

Reviewers #1 and #3 approved the manuscript without new suggestions. We address here the latest suggestions of Reviewer #2. The comments of the reviewer are marked by the letter **C** and our replies by the letter **R**.
but not for particles.

[Redacted]